# Association between blood miR-26a levels following artificial insemination, and pregnancy outcome in dairy cattle

**Thomas Tzelos¤, Seungmee Lee, Alex Pegg, F. Xavier Donadeu** *

The Roslin Institute and Royal (Dick) School of Veterinary Studies, University of Edinburgh, Easter Bush, Midlothian, United Kingdom

¤ Current address: Moredun Research Institute, Pentlands Science Park, Penicuik, United Kingdom
* xavier.donadeu@roslin.ed.ac.uk

**Data Availability Statement:** All relevant data are within the paper and its Supporting Information files.

## Abstract

Early pregnancy diagnosis is key to maximise productivity of dairy herds. We previously showed that an increase in the levels of miR-26 could be detected as soon as day 8 of pregnancy in heifers. The aims of this study were to determine whether 1) plasma miR-26 levels would be distinctly elevated, retrospectively, early after artificial insemination in lactating cows with successful compared to failed pregnancies, 2) the early increase in miRNA levels in cows with successful pregnancy could be accounted for by changes in miRNA expression in white blood cells (WBCs), presumably induced by the effects of embryo-derived interferon tau (IFNt), and 3) plasma miRNA levels may provide a reliable early predictor of pregnancy that could be used at a herd level. Blood samples were taken from a total of 34 dairy cows (lactation number 1 to 4) before (D0) and 9 and 18 days after artificial insemination at oestrus, followed by confirmation of pregnancy status by ultrasound on D32. In addition, WBCs collected from non-pregnant cows (n = 4) were stimulated *in vitro* with recombinant ovine IFNt (0–100 pg/ml). Levels of miRNAs and ISG15, a known IFNt-induced gene, were quantified by qPCR. Relative to D0, a larger increase in plasma miR-26a (P = 0.04) occurred on D9 in cows later confirmed to be pregnant (n = 12) than in cows with a failed pregnancy (n = 22). Expression of miR-26a in WBCs was not affected (P>0.1) by pregnancy status or IFNt stimulation in vitro, in contrast to ISG15 expression which increased markedly (P<0.0001) both in WBC samples collected on D18 from animals later confirmed to be pregnant, and in WBCs after stimulation with IFNt *in vitro*. Finally, ROC analyses revealed that miR-26a on D9 or D18 could predict pregnancy outcome with much lower accuracy than WBC ISG15 on D18 (Likelihood ratio, 2.3 vs 15.4). In summary, a modest increase in plasma miR-26a levels occurs during early pregnancy in mature dairy cows which may not accounted for by changes in miRNA levels in WBCs or the effects of IFNt. Moreover, compared to ISG15, changes in miR-26a levels may not provide an accurate test for early diagnosis of pregnancy in cows.

**Funding:** This study was funded by a Biotechnology and Biological Sciences Research Council award to FXD (BB/T004037/1) The funders had no role in study design, data collection and analysis, decision to publish, or preparation of the manuscript.

**Competing interests:** The authors have declared that no competing interests exist.

## Introduction

Early pregnancy diagnosis is key to maximize reproductive efficiency in precision animal farming. This is particularly important in dairy cattle since pregnancy rates in modern commercial herds continue to be unacceptably low (typically ≤40%), with an average of three services required per pregnancy [1]. Most pregnancy losses in high-producing dairy cows occur during the first three weeks following insemination [2]. Failure to promptly re-inseminate before the next oestrus, i.e. within 21 days post-insemination, leads to extended calving intervals and a significant reduction in overall herd milk production [3]. Critically, even with more recent advances such as electronic monitoring system, detection of oestrus in dairy farms remains overall inefficient [4], and confirmation of pregnancy using available tools such as transrectal ultrasonography or PAG detection in blood or milk is only possible at 24 days or later. As a result, opportunities to re-inseminate cows that fail to become pregnant after service are often missed unnecessarily, with important consequences for reproductive efficiency and milk productivity. Detection of Early Pregnancy Factor, Interferon Stimulated Genes (ISGs) and pregnancy-associated miRNAs have been proposed for pregnancy diagnosis before 21 days in cattle, although the accuracy of such assays for discriminating successful from failed pregnancies has been shown to be limited or, in the case of miRNAs, has not been evaluated properly (reviewed by [5]).

Body fluid miRNAs provide distinct advantages as tissue function biomarkers. Clinical applications using biofluid miRNA levels as diagnostic biomarkers exist or are being developed for a wide array of human diseases, and applications have also actively been explored in animals [6,7], for example, for pregnancy detection in cattle [8–12]. Our group was the first to provide a detailed characterisation of the bovine blood miRNome and to report changes in circulating miRNA profiles associated with early pregnancy in cattle [8]. Using RNA-sequencing in dairy heifers we identified 77 miRNAs which levels changed significantly in plasma during the first 60 days of pregnancy [9]. Of note, a consistent finding in our previous studies was an increase in miR-26a levels during early pregnancy, which could be detected as soon as day 8 after insemination but not after sham-insemination in heifers [8,9]. Whether similar changes in miR-26a also occur during early pregnancy in lactating cows has not been established. Moreover, a distinct increase in levels of a related miRNA, miR-26b, during early pregnancy was separately reported in bovine blood [13] and milk [14] samples, whereas miR-26a levels increased in blood serum of day-16 pregnant sows [15]. These findings suggest a role for the miR-26 family in early pregnancy responses in livestock. However, the relative changes of miR-26a and b during early pregnancy and whether they could be used as an effective test for early discrimination of successful vs. failed pregnancies have not been determined in any species.

Interferon tau provides the embryo signal mediating maternal recognition of pregnancy in ruminants. There is consensus that a systemic response to IFNt can be detected in the dam by day 16 after insemination, as an increase in the expression of ISGs in blood cells, although there is also evidence that responses to IFNt could occur earlier than that [16]. Previous data showed that, although widely distributed across body tissues, miR-26a is expressed at distinctly high levels in blood cells [17,18]. Moreover, the miR-26 family has been reported to be involved in inflammatory and anti-viral responses [19,20], in addition to regulating cell growth and differentiation in cardiovascular [21], neuronal [22], adipose [23] and ovary [24], among other body tissues. This raises the possibility that he observed increase in plasma miR-26 levels during early pregnancy in cattle could be part of an early immune cell response to embryo-derived IFNt.

Thus, the present study was designed to quantify changes in levels of miR-26a and b during early pregnancy in lactating cows to test the hypotheses that 1) plasma miR-26 levels would be

distinctly elevated, retrospectively, early after artificial insemination (at 9 days) in cows with successful compared to failed pregnancies, and 2) this would be associated with higher miR-26 levels in WBCs of pregnant compared to non-pregnant cows, an effect which in turn would be mediated by IFNt. Finally, we wanted to assess the potential of plasma miR-26 quantification to reliably discriminate between successful and failed pregnancies before 21 days post-insemination at a herd level.

## Materials and methods

### Animal sampling

All animal procedures were performed with approval from The Roslin Institute (University of Edinburgh) Animal Welfare and Ethical Review Board and following the UK Animals (Scientific Procedures) Act, 1986. The study was carried out in compliance with the ARRIVE guidelines.

A total of 34 Holstein-Friesian or Holstein-Friesian-cross cows held at the University of Edinburgh's Langhill farm were used over two breeding seasons. All animals were between their 1st and 4th lactation at the time of the study. Cows were oestrus-synchronised followed by artificial insemination (AI) at oestrus as described previously [8]. Blood samples were collected form the tail vein from each animal before (D0) and at 9 and 18 days after AI, and pregnancy detection was performed on Day 32 by trans-rectal ultrasonography.

On each time-point, 10 ml of blood were collected by jugular venepuncture in EDTA coated tubes (Vacutainer Lavender; Scientific Laboratory Supplies, UK), placed on ice and transported to the laboratory within 1 hour of collection. Samples were then processed as described previously [8]. In brief, tubes were centrifuged for 10 minutes at 1,900xg at 4˚C. The plasma was then transferred to clean 1.5ml tubes and centrifuged for 10 minutes at 16,000xg at 4˚C to pellet any contaminating blood cell debris and platelets. The cleared plasma was transferred to new 1.5ml tubes and stored immediately at -80˚C. Additionally, the buffy coat was harvested from the EDTA tubes and transferred to labelled 15 ml tubes. Erythrocytes were lysed by incubation in 5× volume of Red Blood Cell (RBC) lysis buffer (0.144 M ammonium chloride/0.175 M Tris; pH 7.4) for 5 min at room temperature. The white blood cell (WBC) pellet was washed three times in PBS and used immediately for IFNt stimulation or stored in TRI reagent LS (Sigma-Aldrich, USA) at -80˚C.

### Stimulation of WBCs with recombinant ovine IFNt (rIFNt)

WBCs were collected as described above from 4 additional, non-pregnant, lacating cows, and were re-suspended in RPMI containing 10% foetal calf serum and 1% Penicilline/Streptomicine+Glutamine. Cells were counted using a haemocytometer and trypan blue (Sigma-Aldrich, USA), and the RPMI media volume was adjusted to $2x10^6$ cells/ml. Cells harvested from each animal at the above concentration were stimulated in duplicate 24-well clear TC treated flat bottom plates (Corning, USA) with rIFNt diluted in RPMI at a final concentration of 0, 6.25, 25 or 100 pg/ml [25]. Cells were then incubated at 38.5˚C and 5%CO2 for 24h, as described [16], after which they were harvested into a tube, centrifuged at 500xg for 5 minutes at 4˚C and washed twice with PBS. Finally, cells were re-suspended in 1ml of TRI reagent LS (Sigma-Aldrich, USA) and stored at -80˚C until further processing.

### RNA analyses

RNA was extracted from plasma (300μl) or WBC pellets using TRI reagent LS, following the manufacturer's protocol. For plasma samples, an exogenous miRNA control, syn-cel-miR-39-

3p (0.25 fmol; Qiagen, NL), was spiked-in during RNA extraction, and glycogen (180 μg; Sigma-Aldrich, USA) was added to aid with the visualisation of precipitated RNA, which was finally dissolved in 20 μl of RNase-free water. WBC RNA pellets were dissolved in 25 μl of RNase-free water and quantified using a spectrophotometer (Nanodrop ND-1000; Thermo Fisher Scientific, USA). In each case, RNA was used immediately for cDNA synthesis or was frozen at −80˚C.

For miRNA analyses, reverse transcription (10 μl reactions) was performed using 3μl of the plasma RNA sample or 500 ng of WBC RNA, and the miRCURY LNA RT kit (Qiagen, USA), following the manufacturer's protocol. A total of 3μl of the diluted cDNA (1:60) was then used in 10 μl qPCR reactions using miRCURY LNA SYBR Green PCR kit (Qiagen, USA) in an Agilent Mx3005P qPCR system (Agilent Technologies, USA) as per the manufacturer's protocol. Raw fluorescence data were processed using Agilent MxPro software. Relative transcript abundance was obtained by extrapolating Ct values from a standard curve prepared from a sample pool. Primers (miRCURY LNA miRNA PCR Assay, Qiagen, USA) against bta-miR-26a, bta-miR-26b, bta-miR-101, and the normaliser genes, cel-miR-39-3p (for plasma samples) and U6b (for WBC samples), were used. Before PCR analyses, we confirmed the absence of cross-amplification between miR-26a and b primers.

Before mRNA analyses, WBC RNA samples were treated with DNase using the RQ1 kit (Promega, USA) following the manufacturer's protocol. Subsequently, RNA (250 ng) was used for reverse transcription (10μl reactions) using Superscript III (Thermo Fisher, USA). cDNA samples were then diluted 1:300 and used (2μl) in 10 μl qPCR reactions using SensiFast (Bioline, UK) and bovine-specific primers targeting ISG15 (fwd/rev, `GCAGACCAGTTCTGGCTG TCT`/`CCAGCGGGTGCTCATCAT`) and the two normalisers, 18S (`GTGCATGGCCGTTCTTAG TTG`/`AGCATGCCCAGAGTCTCGTT`) and ACTB (`TCACCAACTGGGACGACATG`/`CGTTG TAGAAGGTGTGGTGCC`), in an Agilent Mx3005P qPCR system (Agilent Technologies, USA) as per the manufacturer's protocol. Raw fluorescence data were processed using Agilent MxPro software. Relative transcript abundance was obtained by extrapolating Ct values from a standard curve prepared from a sample pool.

## Statistical analyses

All statistical analyses were performed using Minitab 18 Statistical Software (Minitab, LLC; USA). Data were assessed for normality using the Kolmogorov-Smirnoff test ($P>0.01$) and not normally distributed data were log-transformed before analyses. Data were then analysed using two-way ANOVA to test for main effects of group, experimental day and their interaction using Animal and Lactation number as covariates. If significant, this was followed by a post-hoc Bonferroni test. Finally, bta-miR-26a and ISG15 expression data were tested with Receiver Operation Characteristic (ROC) curves for its ability to distinguish between pregnant and non-pregnant groups. ROC curve analyses were performed using GraphPad Prism 9 (GraphPad software, San Diego, CA). In all cases, significance was considered at $P< 0.05$.

## Results

### Effects of pregnancy on plasma miRNA levels

Based on our previous finding of a distinct increase in plasma miR-26 during early pregnancy in dairy heifers [9], we first investigated whether levels of miR-26a and b after AI would be distinctly higher, retrospectively, in mature dairy cows with successful *vs* failed pregnancies. For reference, miR-101 was included in our analyses, as this miRNA was also increased during early pregnancy in heifers (9). Out of 34 cows inseminated at oestrus (D0), 12 were diagnosed as pregnant (P) and 22 as not pregnant (NP) by ultrasonography on D32. Because baseline

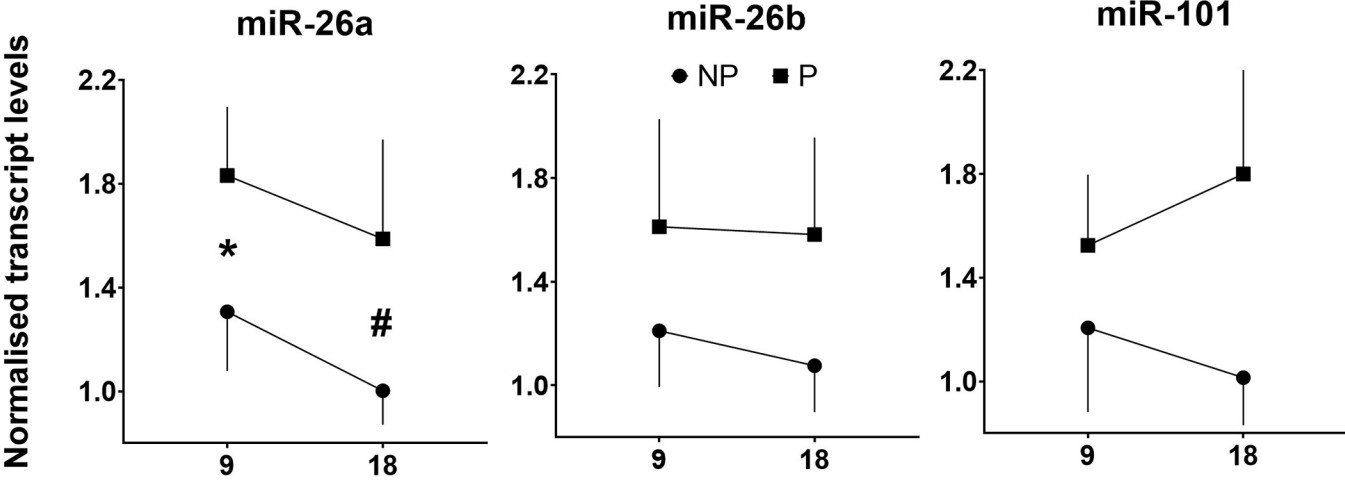

**Fig 1. Relative transcript levels (Mean + SE, normalised to cel-miR-39 levels) of miR-26a, miR-26b and miR-101 in plasma collected on Days 9 and 18 days after AI from mature dairy cows that were later (day 32) confirmed by ultrasound to be pregnant (P) or non-pregnant (NP).** For each miRNA and animal, expression levels on Days 9 and 18 were normalised to Day 0 levels (represented by 1.0 in the y axis). Significant differences between groups within a day are shown by * (P<0.05) or # (P<0.1).

miRNA levels on D0 were different between the two groups of animals (miR-26b and miR-101, P<0.05), D9 and D18 miRNA values were normalised to Day 0 values within each animal before analyses (Fig 1). This revealed significant main effects of Group for miR-26a (P = 0.007) and, more modestly, miR-101 (P = 0.03). Moreover, as shown by post-hoc testing, significant differences between groups at individual time-points were detected for miR-26a only, specifically, higher miRNA levels in P than NP cows on D9 (P = 0.04). Based on this, we focused all subsequent analyses on miR-26a.

### Effects of pregnancy and rIFNt stimulation on levels of miR-26a and ISG15 in white blood cells (WBCs)

Given the reported involvement of miR-26a in immune processes [19,26] we then wanted to determine whether the observed increase in plasma miR-26a levels during early pregnancy in cattle may derive from an increase in miRNA expression by blood cells, presumably as a response to embryo-derived IFNt. To do this, we compared the expression of miR-26a in WBCs between P and NP cows on D9 and D18, and also in bovine WBCs after stimulation with different doses of rIFNt *in vitro*. As shown in Fig 2A, expression of miR-26 was not different (P>0.1) between P and NP on either day. In contrast, ISG15 expression was much higher (>5-fold) in P than NP cows on D18 (P<0.0001). Similarly, stimulation of WBCs with rIFNt induced a robust dose-dependent increase in ISG15 levels (P<0.0001) but did not affect miR-26a expression.

### Plasma miR-26a levels as an early predictor of pregnancy in cattle

Finally, we sought to establish the value of plasma miR-26a levels as a non-invasive, early predictor of pregnancy in mature cows. To that end, we performed Receiver Operating Characteristic (ROC) curve analyses (Fig 3) using miR-26 data as above from P and NP cows on each of D9 and D18, and we compared this to curves obtained with WBC ISG15 data on the same days. Best performance data, as indicated by the highest combined values for sensitivity and

a)

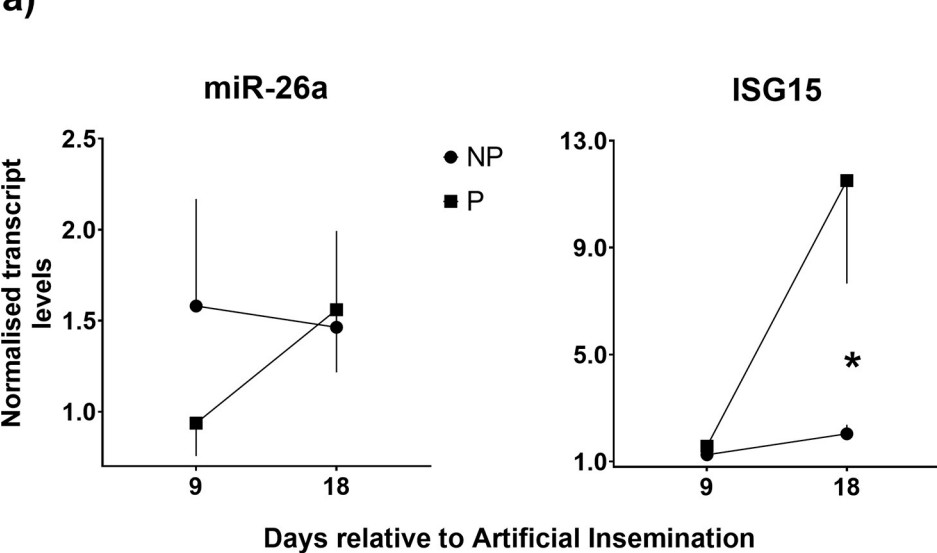

b)

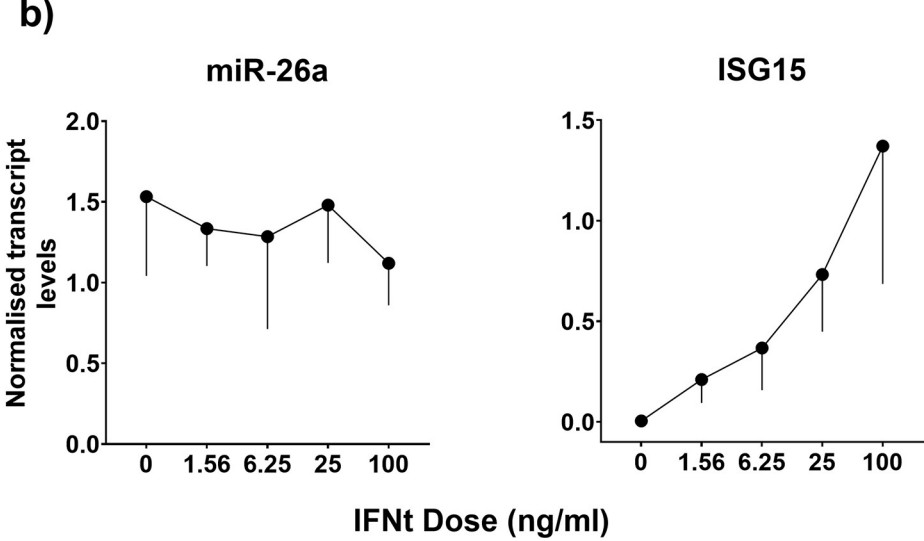

**Fig 2. Relative transcript levels (Mean + SE) of miR-26a (normalised to cel-miR-39 levels) and ISG15 (normalised to 18S and ACTB).** A) Transcript levels in plasma collected on Days 9 and 18 days after AI from mature dairy cows that were later (day 32) confirmed by ultrasound to be pregnant (P) or non-pregnant (NP). For each miRNA and animal, expression levels on Days 9 and 18 were normalised to Day 0 levels (represented by 1.0 in the y axis). B) Transcript levels in bovine WBCs after stimulation with increasing doses of rIFNt. Significant effects were detected for ISG15 levels in plasma (Group x Day, P<0.0001) and after IFNt stimulation (Dose, P<0.0001). Differences between groups within a day are shown by * (P<0.05).

specificity for each target and day, are shown in Table 1. As expected, plasma miR-26a performed better on D9 than D18 although both performed considerably worse than WBC ISG15 on D18, as indicated by sensitivity/specificity values (%) of 83.3/63.6 and 85.7/94.4 for miR-26a on D9 and ISG15 on D18, respectively.

## Plasma miR-26a Day 9

## WBC ISG15 Day 18

**Fig 3. ROC curves obtained using plasma miR26a (Day 9) and WBC ISG15 (Day 18) data (normalised to D0 values for each animal) from dairy cows later diagnosed as pregnant or non-pregnant (Day 32).**

## Discussion

Given their demonstrated potential as non-invasive, relatively stable, diagnostic biomarkers, there has been significant interest in the use of biofluid miRNAs in veterinary diagnostics, and more specifically, for early pregnancy prediction in cattle. Since our original report identifying an association between increased plasma miR-26 levels and early pregnancy [8], numerous studies have investigated the association between blood miRNA profiles and pregnancy outcome following AI in cattle [10–12]. Although differences in multiple miRNAs were reported, there has been little consistency in results among the various studies. This is likely due to the use of different sample types (blood-derived extracellular vesicles *vs* serum *vs* plasma), experimental models (natural *vs* surrogate pregnancy), breed (dairy *vs* beef), and in some cases low sample numbers, or even lack of validation of the high-throughput data obtained. However, several independent studies in addition to our earlier work reported an association between biofluid levels of miR-26 family miRNAs and early pregnancy [13,14,27]. In the present study, the hypothesis that a distinct increase in miR-26a occurs soon after artificial insemination (Day 9) in cows that will successfully establish a pregnancy compared with those with failed

**Table 1. Threshold, sensitivity and specificity values obtained from ROC curve analyses of plasma miR-26a (D9 and D18) and WBC ISG15 (D18) data (normalised to D0 values for each animal).** Highest combined values of sensitivity and specificity obtained for each target are shown.

| Target | Threshold | Sensitivity % | 95% CI | Specificity % | 95% CI | Likelihood ratio |
|---|---|---|---|---|---|---|
| miR-26a D9 | > 1.360 | 83.33 | 55.20% to 97.04% | 63.64 | 42.95% to 80.27% | 2.292 |
| miR-26a D18 | > 0.8300 | 83.33 | 55.20% to 97.04% | 50.00 | 30.72% to 69.28% | 1.667 |
| ISG15 D9 | > 1.290 | 75.00 | 40.93% to 95.56% | 72.22 | 49.13% to 87.50% | 2.700 |
| ISG15 D18 | > 4.665 | 85.71 | 48.69% to 99.27% | 94.44 | 74.24% to 99.72% | 15.43 |

pregnancies, was confirmed. This novel result expands on our earlier findings of an increase in miR-26 levels during early pregnancy in heifers [9]. Yet, the changes observed in miR-26a levels were only modest, and ROC analyses confirmed that they may not provide a reliable predictor of pregnancy, as demonstrated by reduced assay performance compared with WBC ISG levels on Day 18, which provided much higher specificity/sensitivity values consistent with those reported in earlier studies [28,29]. Moreover, from a managerial point of view, the need to collect an additional sample before insemination (D0) in order to use fold changes rather than non-transformed miR-26 levels for pregnancy diagnosis, may not be practical and further limit its value as a predictive biomarker for dairy cattle. It is worth noting that, in our study, embryonic loss in cows in the non-pregnant group could have occurred at any time between AI and Day 32 (when diagnostic ultrasonography was performed), with the highest incidence expected between Days 8 and 16 [2]. Thus, the levels of miRNAs obtained may have been variably affected by the presence or absence of a healthy embryo at the time of sampling, a possibility that should be explored in future studies involving collection of embryos at different times after AI in dairy cows. Moreover, previous studies [8–9] have shown plasma to be an optimum substrate for robustly quantifying miR-26 levels in blood in cattle, however, the use of alternative fractions, for example serum or extracellular vesicles, for assessing associations with successful pregnancy could also be explored in the future. Regardless, we conclude from the results above that, despite its initial promise, miRNA levels do not provide at present a useful tool for early pregnancy diagnosis (before Day 21) in cattle.

The second hypothesis that an increase in plasma miRNA levels would be associated with an increase in miRNA expression in WBCs of early pregnant cows, possibly induced by IFNt, was not supported. This was in spite of a distinct increase in levels of the known IFNt target, ISG15, both in WBCs from early pregnant cows and in response to *in vitro* stimulation of WBCs with increasing doses of rIFNt. This result was somewhat unexpected given that, albeit ubiquitous, miR-26a is expressed at relatively high levels in different types of leukocytes compared to other body cells and tissues [17,30], and moreover miR-26a has been implicated in immune processes including regulation of type I interferon responses during viral infection in pigs [31] and attenuation of immune gene activation during bovine implantation [26]. Although unlikely, we cannot rule out that the observed increase in plasma levels of miR-26a during early pregnancy may be accounted for by elevated expression of this miRNA in a single or a few immune cell subsets only, which may be not be readily detectable by analysing unfractioned WBC samples. However, the present results suggest that increased levels of miR-26a in plasma during early pregnancy originate from tissues other than blood, and that miR-26a is not induced by embryo-derived IFNt.

The results of recent studies raise the intriguing possibility that the increase in miR-26 levels in the circulation of early pregnant cows may, at least in part, derive from production by the embryo itself or the pregnant uterus. Thus, it has been shown that miR-26 family miRNAs are actively expressed by bovine and pig embryos [18,27], and that the levels of these miRNAs in intrauterine extracellular vesicles increase dramatically by the end of week three of pregnancy in cattle [26]. These changes may presumably reflect an involvement of miR-26 in regulating development of the early conceptus [32] and/or in down-regulation of immune-related genes to facilitate implantation [26]. Regardless of the biological function of miR-26 during early pregnancy, an increase in intra-uterine production of miR-26a during the first two weeks of pregnancy could explain the changes in levels of this miRNA in blood observed on Day 8 to 9 after AI in the present and previous studies [9]. This hypothesis should be tested in future studies by determining changes in miR-26a in the bovine embryo and pregnant uterus at earlier time-points than reported by Nakamura et al. [26], namely, during the first two weeks after AI.

## Conclusion

In summary, we have shown that plasma levels of miR-26a on Day 9 after AI distinctly increase in dairy cows that will go on to establish a pregnancy compared to cows with a failed pregnancy, thus expanding on our previous findings in heifers. Moreover, our results suggest that this increase in plasma miR-26a levels does not result from an increase in miR-26a expression in blood cells and also that it is not induced in response to embryo-derived IFNt. We conclude that the modest increase in miR-26a levels in plasma may not provide a reliable predictor of early pregnancy in dairy cattle.

## Supporting information

**S1 File.**
(XLSX)

## Acknowledgments

We are thankful to staff at Langhill Farm, particularly Mr Wilson Lee, for carrying out animal work. We are also extremely grateful to Dr. Claire Stenhouse and Dr. Fuller W. Bazer from Texas A&M University for the gift of recombinant ovine IFNt.

## Author Contributions

**Conceptualization:** Thomas Tzelos, F. Xavier Donadeu.

**Formal analysis:** Thomas Tzelos, Seungmee Lee, Alex Pegg.

**Funding acquisition:** F. Xavier Donadeu.

**Investigation:** Thomas Tzelos, Seungmee Lee, Alex Pegg.

**Methodology:** Thomas Tzelos, Seungmee Lee, Alex Pegg.

**Project administration:** F. Xavier Donadeu.

**Resources:** F. Xavier Donadeu.

**Validation:** Thomas Tzelos, Alex Pegg.

**Writing – original draft:** F. Xavier Donadeu.

**Writing – review & editing:** Thomas Tzelos, Seungmee Lee, Alex Pegg, F. Xavier Donadeu.

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
