## [Decision Letter · Decision Letter 0]

8 Jun 2023

PONE-D-23-07566Association between blood miR-26a levels following artificial insemination and pregnancy outcome in dairy cattlePLOS ONE

Dear Dr. Donadeu,

Thank you for submitting your manuscript to PLOS ONE. After careful consideration, we feel that it has merit but does not fully meet PLOS ONE’s publication criteria as it currently stands. Therefore, we invite you to submit a revised version of the manuscript that addresses the points raised during the review process.

We look forward to receiving your revised manuscript.

Kind regards,

Muhammad Mazhar Ayaz, Ph.D

Academic Editor

PLOS ONE

“This study was funded by a Biotechnology and Biological Sciences Research Council award to FXD (BB/T004037/1). We are thankful to staff at Langhill Farm, particularly Mr Wilson Lee, for carrying out animal work. We are also extremely grateful to Dr. Claire Stenhouse and Dr. Fuller W. Bazer from Texas A&M University for the gift of recombinant ovine IFNt.”

“FXD

BBSRC

BB/T004037/1

https://www.ukri.org/councils/bbsrc/

The funders had no role in study design, data collection and analysis, decision to publish, or preparation of the manuscript”

Reviewers' comments:

Reviewer's Responses to Questions

**Comments to the Author**

1. Is the manuscript technically sound, and do the data support the conclusions?

Reviewer #1: Yes

Reviewer #2: Yes

2. Has the statistical analysis been performed appropriately and rigorously? 

Reviewer #1: Yes

Reviewer #2: Yes

3. Have the authors made all data underlying the findings in their manuscript fully available?

Reviewer #1: Yes

Reviewer #2: Yes

4. Is the manuscript presented in an intelligible fashion and written in standard English?

Reviewer #1: Yes

Reviewer #2: Yes

5. Review Comments to the Author

Reviewer #1: Line 44-45: “Critically, detection of oestrus in dairy farms remains overall inefficient (4)”. This claim is supported by a 20+ year reference. Significant advances in the detection of estrus in bovine females have been made in recent years. For example there are several electronic monitoring systems that improve estrus detection. Use a more recent reference and indicate that even with the evolution of recent years, estrus detection remains a limiting factor for reproductive efficiency.

Lines 47-49: “As a result, opportunities to re-inseminate cows that fail to become pregnant at first or second service are often missed unnecessarily, with important consequences for reproductive efficiency and milk productivity”. Not just the first and second, but after every insemination.

Line 236 “easy to quantify”: What would be the proper definition of "easy to quantify"?. I believe that in this specific case, this is not the case, in view of the complexity described in the material and methods and the authors' description in the discussion text.

Reviewer #2: This is an interesting study evaluating the dynamics of miRNA-26 in blood in pregnant versus non pregnant lactating cows. This is a topic of interest as efforts are continuously made in order to diagnose pregnancy in cattle before 21 days after AI. Although results presented in this manuscript are partly negative, they are completely worth presenting.

I would have some comments to be addressed by the authors before the manscript is accepted for publication:

Abstract: ISG15 is occuring towards the end of the abstract from "nowhere". i consider it is worth introducing this factor before, otherwise it is confusing for the reader.

Introduction: what is known about physiological role of miRNA-26?

Material and Methods: I appreciate the detailed presentation of the RNA analyses by the authors but I find the cell culture part very short and lacking important details. In the way it is presented at the moment, I am not sure if it can be succesfully reproduced. Which type of plates were used? How long? How many samples? How were the cells collected and processed until storage.

Discussion: this part is a little shallow and does not discuss all aspects possibly involved. For example, authors state in the introduction, that most pregnancy losses occur during the first 3 weeks after AI. How can this have affected your results in this study? Also, could it be that plasma is not the ideal sample for detecting changes in miRNA-26 expression?

6. PLOS authors have the option to publish the peer review history of their article (what does this mean?). If published, this will include your full peer review and any attached files.

Reviewer #1: **Yes: **Carlos Antônio de Carvalho Fernandes

Reviewer #2: **Yes: **Dragos Scarlet

---

## [Author Response · Author response to Decision Letter 0]

12 Jun 2023

Reviewer #1: Line 44-45: “Critically, detection of oestrus in dairy farms remains overall inefficient (4)”. This claim is supported by a 20+ year reference. Significant advances in the detection of estrus in bovine females have been made in recent years. For example there are several electronic monitoring systems that improve estrus detection. Use a more recent reference and indicate that even with the evolution of recent years, estrus detection remains a limiting factor for reproductive efficiency.

The text has been amendment and a new reference [4] added as requested

Lines 47-49: “As a result, opportunities to re-inseminate cows that fail to become pregnant at first or second service are often missed unnecessarily, with important consequences for reproductive efficiency and milk productivity”. Not just the first and second, but after every insemination.

Thanks for pointing this out. We have removed ‘at first or second service’ by ‘after service’.

Line 236 “easy to quantify”: What would be the proper definition of "easy to quantify"?. I believe that in this specific case, this is not the case, in view of the complexity described in the material and methods and the authors' description in the discussion text.

Thanks for highlighting this. The reviewer is correct that ‘easy to quantify’ is not the right term. It may be correct in a research context but not in relation to farm animal diagnostics. A more accurate descriptor is ‘relatively stable’ and we have added this to the text. 

Reviewer #2: This is an interesting study evaluating the dynamics of miRNA-26 in blood in pregnant versus non pregnant lactating cows. This is a topic of interest as efforts are continuously made in order to diagnose pregnancy in cattle before 21 days after AI. Although results presented in this manuscript are partly negative, they are completely worth presenting.

Thank you very much for your positive feedback

Abstract: ISG15 is occuring towards the end of the abstract from "nowhere". i consider it is worth introducing this factor before, otherwise it is confusing for the reader.

Thanks for pointing this out. An introduction to ISG15 has been provided in Line 24

Introduction: what is known about physiological role of miRNA-26?

In the original submission (Line 77-78) we referred to miR-26’s involvement in inflammatory and anti-viral responses. We understand the reviewer wishes that we expand on this and we have done so by mentioning additional known roles for miR-26, with references (Lines 79-82). 

Material and Methods: I appreciate the detailed presentation of the RNA analyses by the authors but I find the cell culture part very short and lacking important details. In the way it is presented at the moment, I am not sure if it can be succesfully reproduced. Which type of plates were used? How long? How many samples? How were the cells collected and processed until storage.

Thanks for your observation. Additional details have been provided as requested (lines 122-129)

Discussion: this part is a little shallow and does not discuss all aspects possibly involved. For example, authors state in the introduction, that most pregnancy losses occur during the first 3 weeks after AI. How can this have affected your results in this study? Also, could it be that plasma is not the ideal sample for detecting changes in miRNA-26 expression?

Discussion of the implications of early pregnancy loss on our data as well as the use of plasma has been included (lines 264-273).

---

## [Editor Report · Decision Letter 1]

18 Jul 2023

Association between blood miR-26a levels following artificial insemination, and pregnancy outcome in dairy cattle

PONE-D-23-07566R1

Dear Dr. Donadeu,

We’re pleased to inform you that your manuscript has been judged scientifically suitable for publication and will be formally accepted for publication once it meets all outstanding technical requirements.

Kind regards,

Muhammad Mazhar Ayaz, Ph.D

Academic Editor

PLOS ONE
---

## [Editor Report · Acceptance letter]

4 Aug 2023

PONE-D-23-07566R1 

Association between blood miR-26a levels following artificial insemination, and pregnancy outcome in dairy cattle 

Dear Dr. Donadeu:

I'm pleased to inform you that your manuscript has been deemed suitable for publication in PLOS ONE. Congratulations! Your manuscript is now with our production department. 

Kind regards, 

on behalf of

Dr. Muhammad Mazhar Ayaz 

Academic Editor

PLOS ONE